# DEEP LEARNING IS ROBUST TO MASSIVE LABEL NOISE

## ABSTRACT

Deep neural networks trained on large supervised datasets have led to impressive results in recent years. However, since well-annotated datasets can be prohibitively expensive and time-consuming to collect, recent work has explored the use of larger but noisy datasets that can be more easily obtained. In this paper, we investigate the behavior of deep neural networks on training sets with massively noisy labels. We show on multiple datasets such as MINST, CIFAR-10 and ImageNet that successful learning is possible even with an essentially arbitrary amount of noise. For example, on MNIST we find that accuracy of above 90 percent is still attainable even when the dataset has been diluted with 100 noisy examples for each clean example. Such behavior holds across multiple patterns of label noise, even when noisy labels are biased towards confusing classes. Further, we show how the required dataset size for successful training increases with higher label noise. Finally, we present simple actionable techniques for improving learning in the regime of high label noise.

## 1 INTRODUCTION

Deep learning has proven to be powerful for a wide range of problems, from image classification to machine translation. Typically, deep neural networks are trained using supervised learning on large, carefully annotated datasets. However, the need for such datasets restricts the space of problems that can be addressed. This has led to a proliferation of deep learning results on the same tasks using the same well-known datasets. Carefully annotated data is difficult to obtain, especially for classification tasks with large numbers of classes (requiring extensive annotation) or with fine-grained classes (requiring skilled annotation). Thus, annotation can be expensive and, for tasks requiring expert knowledge, may simply be unattainable at scale.

To address this limitation, other training paradigms have been investigated to alleviate the need for expensive annotations, such as unsupervised learning (Le, 2013), self-supervised learning (Pinto et al., 2016; Wang & Gupta, 2015) and learning from noisy annotations (Joulin et al., 2016; Natarajan et al., 2013; Veit et al., 2017). Very large datasets (e.g., Krasin et al. (2016); Thomee et al. (2016)) can often be attained, for example from web sources, with partial or unreliable annotation. This can allow neural networks to be trained on a much wider variety of tasks or classes and with less manual effort. The good performance obtained from these large noisy datasets indicates that deep learning approaches can tolerate modest amounts of noise in the training set.

In this work, we take this trend to an extreme, and consider the performance of deep neural networks under extremely low label reliability, only slightly above chance. We envision a future in which arbitrarily large amounts of data will easily be obtained, but in which labels come without any guarantee of validity and may merely be biased towards the correct distribution.

The key takeaways from this paper may be summarized as follows:

- **Deep neural networks are able to learn from data that has been diluted by an arbitrary amount of noise.** We demonstrate that standard deep neural networks still perform well even on training sets in which label accuracy is as low as 1 percent above chance. On MNIST, for example, performance still exceeds 90 percent even with this level of label noise (see Figure 1). This behavior holds, to varying extents, across datasets as well as patterns of label noise, including when noisy labels are biased towards confused classes.

- **A sufficiently large training set can accommodate a wide range of noise levels.** We find that the minimum dataset size required for effective training increases with the noise level. A large enough training set can accommodate a wide range of noise levels. Increasing the dataset size further, however, does not appreciably increase accuracy.

- **Adjusting batch size and learning rate can allow conventional neural networks to operate in the regime of very high label noise.** We find that label noise reduces the effective batch size, as noisy labels roughly cancel out and only a small learning signal remains. We show that dataset noise can be partly compensated for by larger batch sizes and by scaling the learning rate with the effective batch size.

## 2 RELATED WORK

**Learning from noisy data.** Several studies have investigated the impact of noisy datasets on machine classifiers. Approaches to learn from noisy data can generally be categorized into two groups: In the first group, approaches aim to learn directly from noisy labels and focus on noise-robust algorithms, e.g., Beigman & Klebanov (2009); Guan et al. (2017); Joulin et al. (2016); Krause et al. (2016); Manwani & Sastry (2013); Misra et al. (2016); Van Horn et al. (2015). The second group comprises mostly label-cleansing methods that aim to remove or correct mislabeled data, e.g., Brodley & Friedl (1999). Methods in this group frequently face the challenge of disambiguating between mislabeled and hard training examples. To address this challenge, they often use semi-supervised approaches by combining noisy data with a small set of clean labels (Zhu, 2005). Some approaches model the label noise as conditionally independent from the input image (Natarajan et al., 2013; Sukhbaatar et al., 2014) and some propose image-conditional noise models (Veit et al., 2017; Xiao et al., 2015). Our work differs from these approaches in that we do not aim to clean the training dataset or propose new noise-robust training algorithms. Instead, we study the behavior of standard neural network training procedures in settings with massive label noise. We show that even without explicit cleaning or noise-robust algorithms, neural networks can learn from data that has been diluted by an arbitrary amount of label noise.

**Analyzing the robustness of neural networks.** Several investigative studies aim to improve our understanding of convolutional neural networks. One particular stream of research in this space seeks to investigate neural networks by analyzing their robustness. For example, Veit et al. (2016) show that network architectures with residual connections have a high redundancy in terms of parameters and are robust to the deletion of multiple complete layers during test time. Further, Szegedy et al. (2014) investigate the robustness of neural networks to adversarial examples. They show that even for fully trained networks, small changes in the input can lead to large changes in the output and thus misclassification. In contrast, we are focusing on non-adversarial noise during training time. Within this stream of research, closest to our work are studies that focus on the impact of noisy training datasets on classification performance (e.g., Sukhbaatar et al. (2014); Van Horn et al. (2015); Zhang et al. (2017)). In these studies an increase in noise is assumed to decrease not only the *proportion* of correct examples, but also their *absolute number*. In contrast to these studies, we separate the effects and show in §4 that a decrease in the number of correct examples is more destructive to learning than an increase in the number of noisy labels.

## 3 LEARNING WITH MASSIVE LABEL NOISE

In this work, we are concerned with scenarios of abundant data of very poor label quality, i.e., the regime in which falsely labeled training examples vastly outnumber correctly labeled examples. In particular, our experiments involve observing the performance of deep neural networks on multiclass classification tasks as label noise is increased.

To formalize the problem, we denote the number of original training examples by $n$. To model the amount of noise, we dilute the dataset by adding $\alpha$ noisy examples to the training set for each original training example. Thus, the total number of noisy labels in the training set is $\alpha n$. Note that by varying the noise level $\alpha$, we do not change the available number of original examples. Thus, even in the presence of high noise, there is still appreciable data to learn from, if we are able to pick it out. This is in contrast to previous work (e.g., Sukhbaatar et al. (2014); Van Horn et al. (2015); Zhang et al. (2017)), in which an increase in noise also implies a decrease in the absolute number

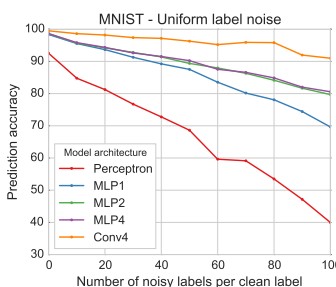 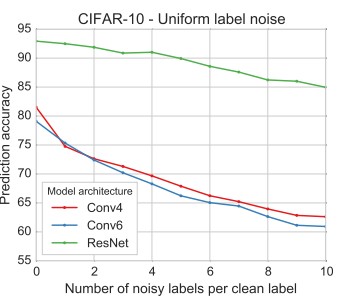 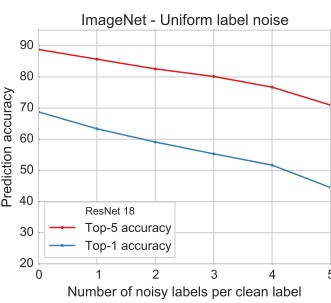

Figure 1: Performance on MNIST as different amounts of noisy labels are added to a fixed training set of clean labels. We compare a perceptron, MLPs with 1, 2, and 4 hidden layers, and a 4-layer ConvNet. Even with 100 noisy labels for every clean label the ConvNet still attains a performance of 91%.

Figure 2: Performance on CIFAR-10 as different amounts of noisy labels are added to a fixed training set of clean labels. We tested ConvNets with 4 and 6 layers, and a ResNet with 101 layers. Even with 10 noisy labels for every clean label the ResNet still attains a performance of 85%.

Figure 3: Performance on ImageNet as different amounts of noisy labels are added to a fixed training set of clean labels. Even with 5 noisy labels for every clean label, the 18-layer ResNet still attains a performance of 70%.

of correct examples. In the following experiments we investigate three different types of noise: uniform label-swapping, structured label-swapping, and out-of-vocabulary examples.

A key assumption in this paper is that unreliable labels are better modeled by an unknown stochastic process rather than by the output of an adversary. This is a natural assumption for data that is pulled from the environment, in which antagonism is not to be expected in the noisy annotation process. Deep neural networks have been shown to be exceedingly brittle to adversarial noise patterns (Szegedy et al., 2014). In this work, we demonstrate that even massive amounts of non-adversarial noise present far less of an impediment to learning.

### 3.1 EXPERIMENT 1: TRAINING WITH UNIFORM LABEL NOISE

As a first experiment, we will show that common training procedures for neural networks are resilient even to settings where correct labels are outnumbered by labels sampled uniformly at random at a ratio of 100 to 1. For this experiment we focus on the task of image classification and work with three commonly used datasets, MNIST (LeCun et al., 1998), CIFAR-10 (Krizhevsky & Hinton, 2009) and ImageNet (Deng et al., 2009).

In Figures 1 and 2 we show the classification performance with varying levels of label noise. For MNIST, we vary the ratio $\alpha$ of randomly labeled examples to cleanly labeled examples from 0 (no noise) to 100 (only 11 out of 101 labels are correct, as compared with 10.1 for pure chance). For the more challenging dataset CIFAR-10, we vary $\alpha$ from 0 to 10. For the most challenging dataset ImageNet, we let $\alpha$ range from 0 to 5. We compare various architectures of neural networks: multi-layer perceptrons with different numbers of hidden layers, convolutional networks (ConvNets) with different numbers of convolutional layers, and residual networks (ResNets) with different numbers of layers (He et al., 2016). We evaluate performance after training on a test dataset that is free from noisy labels. Full details of our experimental setup are provided in §3.4.

Our results show that, remarkably, it is possible to attain over 90 percent accuracy on MNIST, even when there are 100 randomly labeled images for every cleanly labeled example, to attain over 85 percent accuracy on CIFAR-10 with 10 random labels for every clean label, and to attain over 70 percent top-5 accuracy on ImageNet with 5 random labels for every clean label. Thus, in this high-noise regime, deep networks are able not merely to perform above chance, but to attain accuracies that would be respectable even without noise.

Further, we observe from Figures 1 and 2 that larger neural network architectures tend also to be more robust to label noise. On MNIST, the performance of a perceptron decays rapidly with in-

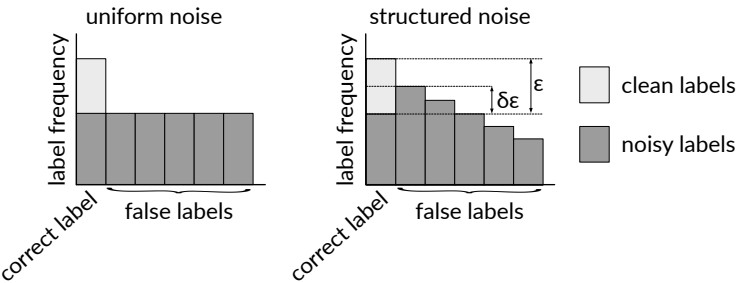

Figure 4: Illustration of uniform and structured noise models. In the case of structured noise, the order of false labels is important; we tested decreasing order of confusion, increasing order of confusion, and random order. The parameter $\delta$ parameterizes the degree of structure in the noise. It defines how much more likely the second most likely class is over chance.

creasing noise (though it still attains 40 percent accuracy, well above chance, at $\alpha = 100$). The performance of a multilayer perceptron drops off more slowly, and the ConvNet is even more robust. Likewise, for CIFAR-10, the accuracy of the residual network drops more slowly than that of the smaller ConvNets. This observation provides further support for the effectiveness of ConvNets and ResNets in particular for applications where noise tolerance may be important.

## 3.2 EXPERIMENT 2: TRAINING WITH STRUCTURED LABEL NOISE

We have seen that neural networks are extremely robust to uniform label noise. However, label noise in datasets gathered from a natural environment is unlikely to follow a perfectly uniform distribution. In this experiment, we investigate the effects of various forms of structured noise on the performance of neural networks. Figure 4 illustrates the procedure used to model noise structure.

In the uniform noise setting, as illustrated on the left side of Figure 4, correct labels are more likely than any individual false label. However, overall false labels vastly outnumber correct labels. We denote the likelihood over chance for a label to be correct as $\epsilon$. Note that $\epsilon = 1/(1 + \alpha)$, where $\alpha$ is the ratio of noisy labels to certainly correct labels. To induce structure in the noise, we bias noisy labels to certain classes. We introduce the parameter $\delta$ to parameterize the degree of structure in the noise. It defines how much more likely the second most likely class is over chance. With $\delta = 0$ the noise is uniform, whereas for $\delta = 1$ the second most likely class is equally likely as the correct class. The likelihood for the remaining classes is scaled linearly, as illustrated in Figure 4 on the right. We investigate three different setups for structured noise: labels biased towards easily confused classes, towards hardly confused classes and towards random classes.

Figure 5 shows the results on MNIST for the three different types of structured noise, as $\delta$ varies from 0 to 1. In this experiment, we train 4-layer ConvNets on a dataset that is diluted with 20 noisy labels for each clean label. We vary the order of false labels so that, besides the correct class, labels are assigned most frequently to (1) those most often confused with the correct class, (2) those least often confused with it, and (3) in a random order. We determine commonly confused labels by training the network repeatedly on a small subset of MNIST and observing the errors it makes on a test set.

The results show that deep neural nets are robust even to structured noise, as long as the correct label remains the most likely by at least a small margin. Generally, we do not observe large differences between the different models of noise structure, only that bias towards random classes seems to hurt the performance a little more than bias towards confused classes. This result might help explain why we often observe quite good results from real world noisy datasets, where label noise is more likely to be biased towards related and confusing classes.

## 3.3 EXPERIMENT 3: SOURCE OF NOISY LABELS

In the preceding experiments, we diluted the training sets with noisy examples drawn from the same dataset; i.e., falsely labeled examples were images from within other categories of the dataset. In

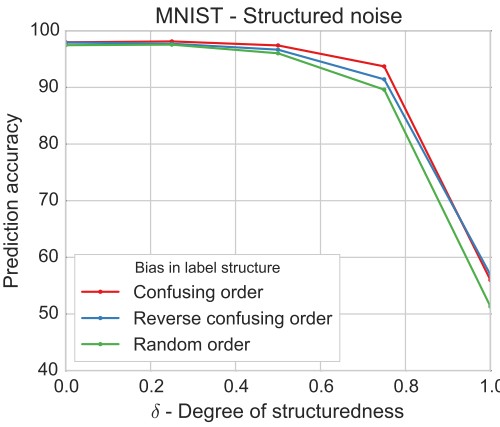 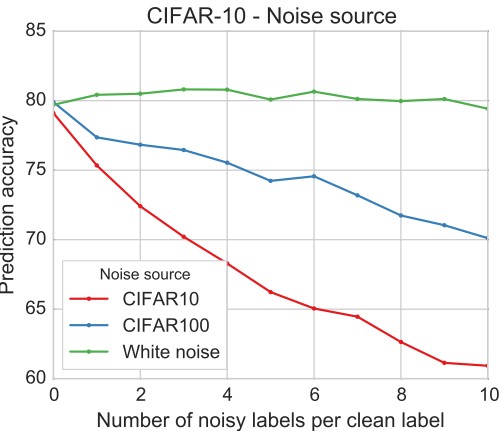

Figure 5: Performance on MNIST with fixed $\alpha = 20$ noisy labels per clean label. Noise is drawn from three types of structured distribution: (1) "confusing order" (highest probability for the most confusing label), (2) "reverse confusing order", and (3) random order. We interpolate between uniform noise, $\delta = 0$, and noise so highly skewed that the most common false label is as likely as the correct label, $\delta = 1$. Except for $\delta \approx 1$, performance is similar to uniform noise.

Figure 6: Performance on CIFAR-10 for varying amounts of noisy labels. Noisy training examples are drawn from (1) CIFAR-10 itself, but mislabeled uniformly at random, (2) CIFAR-100, with uniformly random labels, and (3) white noise with mean and variance chosen to match those of CIFAR-10. Noise drawn from CIFAR-100 resulted in only half the drop in performance observed with noise from CIFAR-10 itself, while white noise examples did not appreciable affect performance.

natural scenarios, however, noisy examples likely also include categories not included in the dataset that have erroneously been assigned labels within the dataset.

Thus, we now consider two alternative sources for noisy training examples. First, we dilute the training set with examples that are drawn from a similar but different dataset. In particular, we use CIFAR-10 as our training dataset and dilute it with examples from CIFAR-100, assigning each image a category from CIFAR-10 at random. Second, we also consider a dilution of the training set with "examples" that are simply white noise; in this case, we match the mean and variance of pixels within CIFAR-10 and again assign labels uniformly at random.

Figure 6 shows the results obtained by a six-layer ConvNet on the different noise sources for varying levels of noise. We observe that both alternative sources of noise lead to better performance than the noise originating from the same dataset. For noisy examples drawn from CIFAR-100, performance drops only about half as much as when noise originates from CIFAR-10 itself. This trend is consistent across noise levels. For white noise, performance does not drop regardless of noise level; this is in line with prior work that has shown that neural networks are able to fit random input (Zhang et al., 2017). This indicates the scenarios considered in Experiments 1 and 2 represent in some sense a worst case.

In natural scenarios, we may expect massively noisy datasets to fall somewhere in between the cases exemplified by CIFAR-10 and CIFAR-100. That is, some examples will be relevant but mislabeled. However, it is likely that many examples will not be from any classes under consideration and therefore will influence training less negatively. In fact, it is possible that such examples might increase accuracy, if the erroneous labels reflect underlying similarity between the examples in question.

## 3.4 EXPERIMENTAL SETUP

All models are trained with AdaDelta (Zeiler, 2012) as optimizer and a batch size of 128. For each level of label noise we train separate models with different learning rates ranging from 0.01 to 1 and pick the learning rate that results in the best performance. Generally, we observe that the higher the label noise, the lower the optimal learning rate. We investigate this trend in detail in §5.

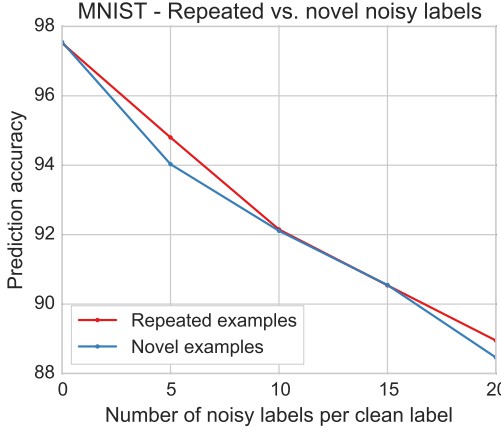 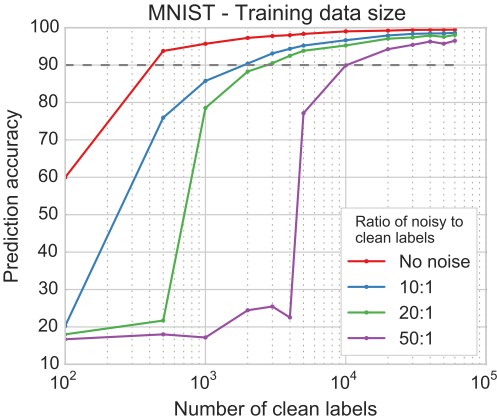

Figure 7: Comparison of the effect of reusing images vs. using novel images as noisy examples. Essentially no difference is observed between the two types of noisy examples, supporting the use of repeated examples in our experiments.

Figure 8: Performance on MNIST at various noise levels, as a function of the number of clean labels. There seems to be a critical amount of clean training data required to successfully train the networks. This threshold increases as the noise level rises. For example, at $\alpha = 10$, 2,000 clean labels are needed to attain 90% performance, while at $\alpha = 50$, 10,000 clean labels are needed.

In Experiments 1 and 2, noisy labels are drawn from the same dataset as the labels guaranteed to be correct. This involves drawing the same example many times from the dataset, giving it the correct label once, and in every other instance picking a random label according to the noise distribution in question. We show in Figure 7 that performance would have been comparable had we been able to draw noisy labels from an extended dataset, instead of repeating images. Specifically, we train a convolutional network on a subset of MNIST, with 2,500 certainly correct labels and with noisy labels drawn either with repetition from this set of 2,500 or without repetition from the remaining examples in the MNIST dataset. The results are essentially identical between repeated and unique examples, supporting our setup in the preceding experiments.

## 4  THE IMPORTANCE OF LARGER DATASETS

Underlying the ability of deep networks to learn from massively noisy data is the size of the data in question. It is well-established, see e.g., Deng et al. (2009), that traditional deep learning relies upon large datasets. We will now see how this is particularly true of noisy datasets.

In Figure 8, we compare the performance of a ConvNet on MNIST as the size of the training set varies. We also show the performance of the same ConvNet trained on MNIST diluted with noisy labels sampled uniformly. We show how the performance of the ConvNet varies with the number of *cleanly labeled training examples*. For example, for the blue curve of $\alpha = 10$ and 1,000 clean labels, the network is trained on 11,000 examples: 1,000 cleanly labeled examples and 10,000 with random labels.

Generally, we observe that independent of the noise level the networks benefit from more data and that, given sufficient data, the networks reach similar results. Further, the results indicate that there seems to be a critical amount of clean training data that is required to successfully train the networks. This critical amount of clean data depends on the noise level; in particular, it increases as the noise level rises. Since performance rapidly levels off past the critical threshold the main requirement for the clean training set is to be of sufficient size.

It is because of the critical amount of required clean data that we have not attempted to train networks for $\alpha \gg 100$. The number of correct examples needed to train such a network might rise above the 60,000 provided in the MNIST dataset. In a real-world dataset, the amount of (noisy) data available

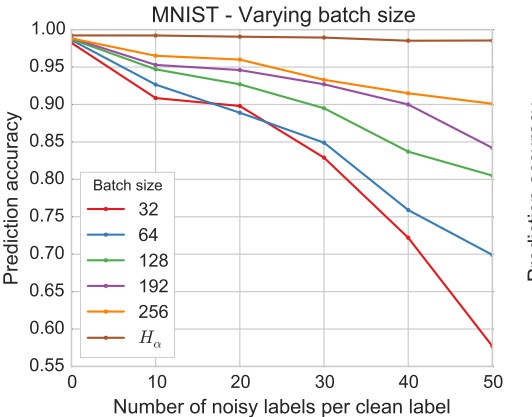
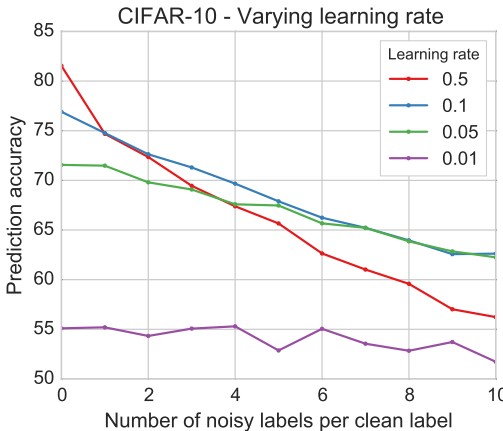

Figure 9: Performance on MNIST for varying batch size as a function of noise level. Higher batch size gives better performance. We approximate the limit of infinite batch size by training without noisy labels, but using the *noisy loss function* $H_\alpha$.

Figure 10: Performance on CIFAR-10 for varying learning rate as a function of noise level. Lower learning rates are generally optimal as the noise level increases.

for training is likely not to be the limiting factor. Rather, considerations such as training time and learning rate may play a more important role, as we discuss in the following section.

## 5 TRAINING ON NOISY DATASETS

In the preceding sections, our results were obtained by training neural networks with fixed batch size and running a parameter search to pick the optimal learning rate. We now look in more detail into how the choice of hyperparameters affects learning on noisy datasets.

### 5.1 BATCH SIZE

First, we investigate the effect of the batch size on the noise robustness of neural network training. In Figure 9, we compare the performance of a simple 2-layer ConvNet on MNIST with increasing noise, as batch size varies from 32 to 256. We observe that increasing the batch size provides greater robustness to noisy labels. One reason for this behavior could be that, within a batch, gradient updates from randomly sampled noisy labels cancel out, while gradients from correct examples that are marginally more frequent sum together and contribute to learning. By this logic, large batch sizes would be more robust to noise since the mean gradient over a larger batch is closer to the gradient for correct labels. All other experiments in this paper are performed with a fixed batch size of 128.

We may also consider the theoretical case of *infinite* batch size, in which gradients are averaged over the entire space of possible inputs at each training step. While this is often impossible to perform in practice, we can simulate such behavior by an auxiliary loss function.

In classification tasks, we are given an input $\mathbf{x}$ and aim to predict the class $f(\mathbf{x}) \in \{1, 2, \ldots, m\}$. The value $f(\mathbf{x})$ is encoded within a neural network by the 1-hot vector $\mathbf{y}(\mathbf{x})$ such that

$$y_k(\mathbf{x}) = \begin{cases} 1 & \text{if } k = f(\mathbf{x}) \\ 0 & \text{otherwise} \end{cases} \tag{1}$$

for $1 \le k \le m$. Then, the standard cross-entropy loss over a batch $X$ is given by:

$$H(X) = -\langle \log \hat{y}_{f(\mathbf{x})} \rangle_X, \tag{2}$$

where $\hat{\mathbf{y}}$ is the predicted vector and $\langle \cdot \rangle_X$ denotes the expected value over the batch $X$. We assume that $\hat{\mathbf{y}}$ is normalized (e.g. by the softmax function) so that the entries sum to 1.

For a training set with noisy labels, we may consider the label $f(\mathbf{x})$ given in the training set to be merely an approximation to the true label $f_0(x)$. Consider the case of $n$ training examples, and $\alpha n$

noisy labels that are sampled uniformly at random from the set $\{1, 2, \ldots, m\}$. Then, $f(\mathbf{x}) = f_0(\mathbf{x})$ with probability $\frac{1}{1+\alpha}$, and otherwise it is $1, 2, \ldots, m$, each with probability $\frac{\alpha}{m(1+\alpha)}$. As batch size increases, the expected value over the batch $X$ is approximated more closely by these probabilities. In the limit of infinite batch size, equation (2) takes the form of a *noisy loss function $H_\alpha$*:

$$H_\alpha(X) := -\frac{1}{1+\alpha} \langle \log \hat{y}_{f_0(\mathbf{x})} \rangle_X - \frac{\alpha}{m(1+\alpha)} \sum_{k=1}^{m} \langle \log \hat{y}_k \rangle_X$$

$$\propto -\langle \log \hat{y}_{f_0(\mathbf{x})} \rangle_X - \alpha \left\langle \log \prod_{k=1}^{m} \hat{y}_k^{1/m} \right\rangle_X \tag{3}$$

We can therefore compare training using the cross-entropy loss with $\alpha n$ noisy labels to training using the noisy loss function $H_\alpha$ without noisy labels. The term on the right-hand side of (3) represents the noise contribution, and is clearly minimized where $\hat{y}_k$ are all equal. As $\alpha$ increases, this contribution is weighted more heavily against $-\langle \log \hat{y}_{f_0(\mathbf{x})} \rangle_X$, which is minimized at $\hat{\mathbf{y}}(\mathbf{x}) = \mathbf{y}(\mathbf{x})$.

We show in Figure 9 the results of training our 2-layer ConvNet on MNIST with the noisy loss function $H_\alpha$, simulating $\alpha n$ noisy labels with infinite batch size. We can observe that the network's accuracy does not decrease as $\alpha$ increases. This can be explained by the observation that an increasing $\alpha$ is merely decreasing the magnitude of the true gradient, rather than altering its direction.

Our observations indicate that increasing noise in the training set *reduces the effective batch size*, as noisy signals roughly cancel out and only small learning signal remains. We show that increasing the batch size is a simple practical means to mitigate the effect of noisy training labels.

## 5.2 LEARNING RATE

It has become common practice in training deep neural networks to scale the learning rate with the batch size. In particular, it has been shown that the smaller the batch size, the lower the optimal learning rate (Krizhevsky, 2014). In our experiments, we have observed that noisy labels reduce the effective batch size. As such, we would expect that lower learning rates perform better than large learning rates as noise increases. Figure 10 shows the performance of a 4-layer ConvNet trained with different learning rates on CIFAR-10 for varying label noise. As expected, we observe that the optimal learning rate decreases as noise increases. For example, the optimal learning rate for the clean dataset is 1, while, with the introduction of noise, this learning rate becomes unstable.

To sum up, we observe that increasing label noise reduces the effective batch size. We have shown that the effect of label noise can be partly counterbalanced for by a larger training batch size. Now, we see that one can additionally scale the learning rate to compensate for any remaining change in effective batch size induced by noisy labels.

## 6 CONCLUSION

In this paper, we have considered the behavior of deep neural networks on training sets with very noisy labels. In a series of experiments, we have demonstrated that learning is robust to an essentially arbitrary amount of label noise, provided that the number of clean labels is sufficiently large. We have further shown that the threshold required for clean labels increases as the noise level does. Finally, we have observed that noisy labels reduce the effective batch size, an effect that can be mitigated by larger batch sizes and downscaling the learning rate.

It is worthy of note that although deep networks appear robust to even high degrees of label noise, clean labels still always perform better than noisy labels, given the same quantity of training data. Further, one still requires expert-vetted test sets for evaluation. Lastly, it is important to reiterate that our studies focus on non-adversarial noise.

Our work suggests numerous directions for future investigation. For example, we are interested in how label-cleaning and semi-supervised methods affect the performance of networks in a high-noise regime. Are such approaches able to lower the threshold for training set size? Finally, it remains to translate the results we present into an actionable trade-off between data annotation and acquisition costs, which can be utilized in real world training pipelines for deep networks on massive noisy data.

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
