# OpenReview forum: "Deep Learning is Robust to Massive Label Noise"
_ICLR.cc/2018/Conference — Reject_

### Official Review · AnonReviewer2 · 2017-11-11
**Bold claims, but in contrast to observations with large-scale real-world data.**

**Rating:** 5
**Confidence:** 4

**Review:**

The paper makes a bold claim, that deep neural networks are robust to arbitrary level of noise. It also implies that this would be true for any type of noise, and support this later claim using experiments on CIFAR and MNIST with three noise types: (1) uniform label noise (2) non-uniform but image-independent label noise, which is named "structured noise", and (3) Samples from out-of-dataset classes. The experiments show robustness to these types of noise.

Review:
The claim made by the paper is overly general, and in my own experience incorrect when considering real-world-noise. This is supported by the literature on "data cleaning" (partially by the authors), a procedure which is widely acknowledged as critical for good object recognition.  While it is true that some image-independent label noise can be alleviated in some datasets, incorrect labels in real world datasets can substantially harm classification accuracy.

It would be interesting to understand the source of the difference between the results in this paper and the more common results (where label noise damages recognition quality). The paper did not get a chance to test these differences, and I can only raise a few hypotheses. First, real-world noise depends on the image and classes in a more structured way. For instance, raters may confuse one bird species from a similar one, when the bird is photographed from a particular angle. This could be tested experimentally, for example by adding incorrect labels for close species using the CUB data for fine-grained bird species recognition.  Another possible reason is that classes in MNIST and CIFAR10 are already very distinctive, so are more robust to noise. Once again, it would be interesting for the paper to study why they achieve robustness to noise while the effect does not hold in general.

Without such an analysis, I feel the paper should not be accepted to ICLR because the way it states its claim may mislead readers.

Other specific comments:
-- Section 3.4 the experimental setup, should clearly state details of the optimization, architecture and hyper parameter search. For example, for Conv4, how many channels at each layer? how was the net initialized? which hyper parameters were tuned and with which values? were hyper parameters tuned on a separate validation set? How was the train/val/test split done, etc. These details are useful for judging technical correctness.
-- Section 4, importance of large datasets. The recent paper by Chen et al (2017) would be relevant here.
-- Figure 8 failed to show for me.
-- Figure 9,10, need to specify which noise model was used.

---

> ### Author Response · Authors · 2018-01-03
> **Response to review**
>
> We would like to thank the reviewer for their time and insightful comments.
>
> We agree with the reviewer that it is key to have clean data to achieve top performance. In our work, we aim to measure precisely the effect of noisy annotations on classification performance, complementing prior work in which increasing noise simultaneously implied a decrease in clean data.
>
> We thank the reviewer for the excellent question, on how our experiments capture the variability of natural, especially fine-grained, datasets. Our use of ImageNet, CIFAR, and MNIST was intended to capture several different paradigms of distinctiveness between classes. We consider noise which is class-dependent but image-independent - that is, two pictures from the same class are equally likely to be mislabeled, with no images being “especially confusing”. Our interest in this formulation was in fact motivated by datasets used for fine-grained identification.
>
> We observed that in the field, practitioners were using large datasets with abundant class-dependent errors, instead of occasional image-dependent errors. For example, a Google Image search for “Blue Jay”, a common species of bird, turns up many incorrect examples. Some are not birds at all (such as the similarly named baseball team). Of those that are, most are high-quality images of other species (such as the Steller’s Jay). We do not observe a prevalence of ambiguous images among those that are falsely labeled.
>
> We have addressed the specific comments in our revision. We are uncertain why Figure 8 should have failed to appear for some readers and have reformatted the image; please let us know if further issues arise.

---

### Official Review · AnonReviewer1 · 2017-11-28
**Learning with Noisy data.**

**Rating:** 4
**Confidence:** 5

**Review:**

The authors study the effect of label noise on classification tasks. They perform experiments of label noise in a uniform setting, structured setting as well provide some heuristics to mitigate the effect of label noise such as changing learning rate or batch size.

Although, the observations are interesting, especially the one on MNIST where the network performs well even with correct labels slightly above chance, the overall contributions are incremental. Most of the observations of label noise such as training with structured noise, importance of larger datasets have already been archived in prior work such as in Sukhbataar et.al. (2014) and Van Horn et. al (2015). Agreed that the authors do a more detailed study on simple MNIST classification, but these insights are not transferable to more challenging domains.

The main limitation of the paper is proposing a principled way to mitigate noise as done in Sukhbataar et.al. (2014), or an actionable trade-off between data acquisition and training schedules.

The authors contend that the way they deal with noise (keeping number of training samples constant) is different from previous setting which use label flips. However, the previous settings can be reinterpreted in the authors setting. I found the formulation of the \alpha to be non-intuitive and confusing at times. The graphs plot number of noisy labels per clean label so a alpha of 100 would imply 1 right label and 100 noisy labels for total 101 labels. In fact, this depends on the task at hand (for MNIST it is 11 clean labels for 101 labels). This can be improved to help readers understand better.

There are several unanswered questions as to how this observation transfers to a semi-supervised or unsupervised setting, and also devise architectures depending on the level of expected noise in the labels.

Overall, I feel the paper is not up to mark and suggest the authors devote using these insights in a more actionable setting.
Missing citation: "Training Deep Neural Networks on Noisy Labels with Bootstrapping", Reed et al.

---

> ### Author Response · Authors · 2018-01-03
> **Response to review**
>
> We would like to thank the reviewer for their time and insightful comments.
>
> The questions of improving training under noisy conditions, cleaning noisy data before training, and reducing noise in dataset acquisition are all very important. As you mention, there has already been excellent work on these questions. Complementary to these lines of research, we aim to show that standard methods can work remarkably well even in the presence of high noise. Certainly, it is always better to have less noise; and if noise is present, then an explicitly noise-robust algorithm might provide better results. However, there are many circumstances in which ConvNets are naively applied without understanding the extent of noise in the data.  For such cases, we believe it is constructive to demonstrate the remarkable robustness of the “vanilla” training paradigm.
>
> We respectfully disagree that prior experiments adequately consider the question of simultaneous large noise and large training set size. In prior work, we have found observations that performance decreases when good training examples are turned into bad ones. This is, however, unsurprising since it essentially represents a double attack: The signal is being diluted with noise *and* there is less total signal. We believe that to understand the robustness to label noise it is key to separate the two effects. To that end, we consider experiments in which the extent of the signal is fixed, but the dilution can be controlled. Specifically, we find that the likely explanation for the previously observed decreases in performance is that the datasets were too small to compensate for the noise.
>
> We did indeed consider alternatives to alpha in parametrizing the extent of noise. As mentioned above, we chose this measurement because it allows us to keep the total amount of signal constant while varying the extent to which this signal is diluted. In light of your helpful comments, we have made some changes in the revision as to how this is described.
>
> We have added the missing citation - thank you for drawing this to our attention.

---

### Official Review · AnonReviewer3 · 2017-12-01
**The paper talks about how various kinds of noise types and levels hurt various deepnets on different problems. Furthermore, the authors give some empirical analysis on how the learning parameters specifically batch size and learning rate should be tweaked i nthe presence of noise.**

**Rating:** 5
**Confidence:** 5

**Review:**

The problem the authors are tackling is extremely relevant to the research community. The paper is well written with considerable number of experiments. While a few conclusions are made in the paper a few things are missing to make even broader conclusions. I think adding those experiments will make the paper stronger!

1. Annotation noise is one of the biggest bottleneck while collecting fully supervised datasets. This noise is mainly driven by lack of clear definitions for each concept (fine-grained, large label dictionary etc.). It would be good to add such type of noise to the datasets and see how the networks perform.
2. While it is interesting to see large capacity networks more resilient to noise I think the paper spends more effort and time on small datasets and smaller models even in the convolutional space. It would be great to add more experiments on the state of the art residual networks and large datasets like ImageNet.
3. Because the analysis is very empirical and authors have a hypothesis that the batch size is effectively smaller when there are large batches with noisy examples it would be good to add some analysis on the gradients to throw more light and make it less empirical.  Batch size and learning rate analysis was very informative but should be done on ResNets and larger datasets to make the paper strong and provide value to the research community.

Overall, with these key things missing the paper falls a bit short making it more suitable for a re submission with further experiments.

---

> ### Author Response · Authors · 2018-01-05
> **Response to review**
>
> We would like to thank the reviewer for their time and insightful comments, and we respond below to the particular issues raised.
>
> 1. We agree with the reviewer that there are different types of noise depending on the granularity and size of the dataset dictionary. This was our motivation in using three different datasets, ImageNet, CIFAR, and MNIST, to capture several different paradigms of distinctiveness between classes.
>
> 2. Our motivation in working with smaller models was greater computational tractability in considering different training set sizes, noise types, and hyperparameter choices. Our experiments with ResNets on ImageNet illustrate the consistency of our main robustness results across models and dataset sizes.
>
> 3. We would be very interested in a theoretical analysis to complement our empirical findings. Establishing our conclusions required a significant amount of experimentation, inducing us to relegate a corresponding theoretical analysis to a separate work.

---

### Public Comment · (anonymous) · 2017-10-31
**Good results, better than in literature**

Here https://arxiv.org/abs/1606.02228 accuracy drops 27%, with 1:1 noisy/clean labels, which is much more than in the paper. Although, may be ResNets are much more robust than CaffeNet.

---

> ### Author Response · Authors · 2017-11-09
> **Response re comparison of results**
>
> Thank you for bringing this up.  Indeed, the difference between ResNets and CaffeNet likely makes a difference in the observed results. Further, in the referred paper, as in other prior work, an increase in noise comes at the expense of a smaller number of clean training examples. As we show in Section 4, this likely causes lower training performance than could be achieved over a large training set with the same level of noise.

---

### Public Comment · ~Michaël_Medeiros_Charbonneau1 · 2017-12-14
**Reproducibility Review for the Reproducibility Challenge**

The extended review we wrote for our machine learning course can be found at https://drive.google.com/file/d/1AHRxuKsR_ywKGAeXUo1QfYbEmdf0Kwxk/view?usp=sharing

Our code can be found at
https://drive.google.com/open?id=1elVTaqdXMDEIQW3qITOYCnKiMYXks_6a

Introduction

We were unable to closely match the authors’ experiments, but had similar trends holding for most, except for the effect of batch size on performance. Their claim that learning is possible under heavy noise was generally confirmed by our team.

The paper having clearly explained plots and noise measures, we focused on trying to reproduce the plots they obtained or, at the very least, confirm the qualitative behavior of neural nets the authors observed for the different datasets and noise settings they selected.

Results

When training a 4-layer convolutional neural net over MNIST with an alpha of 100, the authors of the original paper managed to obtain an accuracy above 90 % when we only managed to reach an accuracy of 20% for such high alpha. This is a large gap, but our experiment still confirms learning beyond random predictions is possible even with such high label noise, as the authors' paper claim.

The paper also explored how many properly labels were needed to get satisfying performance. When using 10 000 correctly labeled images from MNIST with alpha = 20, the authors were able to obtain classification accuracy beyond 90% and our team reached accuracy above 80%. For a fixed alpha, increasing the number of good labels produced an increase in performance in both the authors' attempts and ours for alpha in {0,10,20}. However, for alpha = 50, accuracy drops after 10 000 good labels in our experiment. Moreover, the authors' accuracy takes a large jump around 1000 good labels for alpha=20 when for our reproduction of this experiment, the accuracy increases more smoothly over the whole range for alpha.

For ImageNet, given the size of the original dataset, it was not realistic to expect the same results with our attempt to reproduce the experiment. The results from the original paper for alpha  in {0,1,2,3,4,5} were roughly 70%, 63%, 58%, 55%, 52%, 45% respectively under uniform noise. For alpha increasing from 0 to 5, our classification accuracy obtained decreased from 5% to 0.5%, a much lower accuracy.

The original paper obtained above 60 % accuracy for alpha  when using mislabeled examples from CIFAR-10 whereas we did not manage to obtain accuracy beyond random performance for such high alpha.

Discussion

Here, we summarize what helped reproduce the paper and what made it more difficult.

Noise measures like alpha or delta were properly defined and explained. The nets' types and depths were provided along learning rates and batch sizes for training. However, other hyper-parameters like the number of units per layer or activation function were not specified. As their code was not clearly accessible, we had to implement nets similar to the ones discussed in their paper starting only from the few details shared.

All plots had clear legends and always had an explanation in the text or right below the plot. This made our attempts at reproducing their work much easier. Overall, the paper was well written and organized and its themes and ideas were clearly stated.

In terms of time constraints, we ignore how long it took the authors to complete the experiments they ran. In particular, the number of training epochs required to run any of the neural nets could not be found.

Furthermore, the authors did not mention what preprocessing steps were needed to obtain their results. This is quite important information since preprocessing details state the assumptions made on the data. It is possible that the authors only used raw images for training, but this should have been explicitly stated, as many an alternatives, such as image augmentation, could affect results.

Conclusion

The paper we studied looked at how much noise neural nets can handle when bad labels make up a significant proportion of a training set. Their global claim that neural nets can learn under heavy noise was confirmed by most of our experiments, but matching their classification performances was unsuccessful. More details of implementation would help reproduce their work and confirm the trends their results suggested. Moreover, the time required to complete their study would give a rough initial idea of how long it would take someone to reproduce their work. In our situation, time was limited, so having clear, access to these training settings and extra hyper-parameters would have made our work easier.

Fortunately, the authors' themes and plots were well-explained and understandable. In addition, their metrics and noise definitions were described in details, making their work realistically reproducible.

---

### Decision · Program_Chairs · 2018-01-29
**ICLR 2018 Conference Acceptance Decision**

**Decision:**

Reject

**Comment:**

The paper studies the robustness of deep learning against label noise on MNIST, CIFAR-10 and ImageNet. But the generalization of the claim "deep learning is robust to massive label noise" is still questionable due to the limited noise types investigated.
The paper presents some tricks to improve learning with high label noise (batch size and learning rate), which is not novel enough.